# Species Identity Dominates over Environment in Driving Bacterial Community Assembly in Wild Invasive Leaf Miners

Yu-Xi Zhu,[a] Ya-Wen Chang,[a] Tao Wen,[b] Run Yang,[a] Yu-Cheng Wang,[a] Xin-Yu Wang,[a] Ming-Xing Lu,[a] Yu-Zhou Du[a]

[a]Institute of Applied Entomology, School of Horticulture and Plant Protection, Yangzhou University, Yangzhou, Jiangsu, China

[b]The Key Laboratory of Plant Immunity, Jiangsu Provincial Key Lab for Organic Solid Waste Utilization, Jiangsu Collaborative Innovation Center for Solid Organic Wastes, Educational Ministry Engineering Center of Resource-saving fertilizers, Nanjing Agricultural University, Nanjing, China

**ABSTRACT** The microbiota of invasive animal species may be pivotal to their adaptation and spread, yet the processes driving the assembly and potential sources of host-microbiota remain poorly understood. Here, we characterized microbiota of four *Liriomyza* leaf miner fly species totaling 310 individuals across 43 geographical populations in China and assessed whether the microbiota of the wild leaf miner was acquired from the soil microbiota or the host plant microbiota, using high-throughput 16S rRNA sequencing. Bacterial communities differed significantly among four leaf miner species but did not mirror host phylogeny. Microbiota diversity in the native *L. chinensis* was significantly higher than in three invasive leaf miners (i.e., *L. trifolii*, *L. huidobrensis*, and *L. sativae*), yet the microbial community of the invasive species exhibited a more connected and complex network structure. Structural equation models revealed that host species identity was more important than environmental factors (e.g., geography, climate, or plants) in shaping microbiota composition. Using neutral and null model analyses, we found that deterministic processes like variable selection played a primary role in driving microbial community assembly, with some influence by stochastic processes like drift. The relative degree of these processes governing microbiota was likely correlated with host species but independent of either geographical or climatic factors. Finally, source tracking analysis showed that leaf miners might acquire microbes from their host plant rather than the soil. Our results provide a robust assessment of the ecological processes governing bacterial community assembly and potential sources of microbes in invasive leaf miners.

**IMPORTANCE** The invasion of foreign species, including leaf miners, is a major threat to world biota. Host-associated microbiota may facilitate host adaption and expansion in a variety of ways. Thus, understanding the processes that drive leaf miner microbiota assembly is imperative for better management of invasive species. However, how microbial communities assemble during the leaf miner invasions and how predictable the processes remain unexplored. This work quantitatively deciphers the relative importance of deterministic process and stochastic process in governing the assembly of four leaf miner microbiotas and identifies potential sources of leaf miner-colonizing microbes from the soil-plant-leaf miner continuum. Our study provides new insights into the mechanisms underlying the drive of leaf miner microbiota assembly.

**KEYWORDS** leaf miner, microbial ecology, community assembly, microbial source

Animals harbor diverse microbial communities capable of exerting substantial effects on host physiology, ecology, and evolution (1). Symbiotic microbes may facilitate invasions by insect species that pose a severe threat to environments and economies worldwide (2–4). Insect-associated microbes may provision essential

Address correspondence to Yu-Zhou Du, yzdu@yzu.edu.cn.

The authors declare no conflict of interest.

nutrients or break down difficult to digest plant polymers (5), manipulate host reproduction or influence a range of basic fitness parameters (6, 7), broaden the dietary breadth of herbivores (8–10), and protect hosts from natural enemies and other environmental stress (11–13).

In some cases, invasive insect species may be infected with maternally-transmitted microbes that provide additional heritable genetic variation which may be important to adapting to or persisting in novel or varied environments (14). Heritable symbionts can be obligately or facultatively associated with hosts, but typically live within the body cavity and together should be considered an invasive species complex (4). More common are environmentally-acquired microbes, which typically reside in the insect gut (15). While the microbiota of the founding population may influence invasion success by providing phenotypic plasticity, the novel environment may, in turn, influence the host microbiota and its subsequent function (15, 16). Clarifying the processes driving the composition of invasive insect-associated microbial communities from an evolutionary and ecological perspective is currently an important goal in invasion ecology (17).

Community assemblies rest on four basic processes: diversification, selection, drift, and dispersal (18, 19). Niche theory asserts that deterministic processes, including homogeneous or variable selection, govern community structure, leading communities to converge or diverge, respectively (20). Several deterministic factors, such as host genotype (21), ontogeny (16, 22), diet (23, 24), habitat (25, 26), geographical location (27), and climate (28, 29), as well as microbe–microbe interactions (30, 31) can shape arthropod microbiomes. However, neutral processes like dispersal and drift can also play essential roles in microbial assembly (32–35). Although a growing number of studies have indicated that both deterministic and stochastic processes affect microbiome assembly across diverse systems, including zebrafish (36), cows (37), and humans (38), relatively little is known about the mechanisms structuring microbial communities for most insect systems, including important invasive species. This partly arises from the diversity of insect systems and their associated microbial communities (39), which suggests that the relative importance of various forces may be system specific or vary over time or space. For example, a survey of *Drosophila* microbiomes found that selective factors were most important in shaping the gut communities' structure of flies (40), while honeybee gut communities were mainly governed by stochastic factors (41).

Another current challenge in insect microbiome research is evaluating the transmission dynamics of microbes among multiple trophic networks as well as rapid evolutionary shifts along the mutualist-parasite continuum (42). Microbiome components colonizing the host are expected to be derived from the local environmental pool of microorganisms (15). Often these microbes are sourced from adjacent trophic levels, but they may be derived from any component of the trophic network (43, 44). For instance, investigating the microbiota of the caterpillars, dandelions and soil under greenhouse conditions, Hannula et al. (44) drew the surprising conclusion that foliar-feeding insects primarily acquired their microbiota from the soil rather than the plant.

Leaf mining flies in the genus *Liriomyza* (Diptera: Agromiyzidae) offer an excellent system to investigate the mechanisms underlying microbial community assembly of invasive species. Multiple *Liriomyza* species are notorious globally invasive species, but they can also be efficiently collected in the field and maintained in the laboratory (2, 45). Among this group, three highly polyphagous species, *L. trifolii*, *L. huidobrensis*, and *L. sativae*, which originated in the Americas, have now spread to multiple regions of China where they are causing severe yield losses of numerous ornamental and vegetable crops (46, 47). An oligophagous species *L. chinensis* is a native leaf miner found in a few regions of China (46, 48). To our knowledge, there have been no reports the microbiome of these important plant pests. As such, the environmental sources and composition of the *Liriomyza* microbiome are unknown, and the relative importance of selective and nonselective processes in governing microbiome assembly remains unexplored.

Here, we first characterized the microbiome composition of four leaf miner species consisting of 310 total individuals collected from multiple host plants and locations

across China, using high-throughput 16S rRNA sequencing. Our objectives were to (i) compare the bacterial diversity and composition of native *L. chinensis* relative to three invasive *Liriomyza* species (*L. trifolii*, *L. huidobrensis*, and *L. sativae*), (ii) evaluate the relative contributions of deterministic and stochastic processes to leaf miner microbiota assembly, and (iii) determine the extent to which different environmental factors may impact these effects. We also investigated the microbiota from paired leaf miner samples with their associated food plants and soils to explore the potential sources of leaf miner-colonizing microbes across trophic levels.

## RESULTS

**Bacterial community variation among four host species.** After quality filtering and chimera removal, 112,951,473 reads were obtained from 310 samples, representing 291 wild and 19 laboratory individuals derived from the native leaf miner *L. chinensis*, and the three invasive species *L. trifolii*, *L. huidobrensis*, and *L. sativae* (Fig. S1 and Table S1 in the online supplemental material). The bacterial alpha diversity was significantly different among the four leaf miner species (Wilcoxon statistic: Shannon index: 11.23, $P = 0.0106$; Richness: 9.411, $P = 0.0243$). For all indices tested, the bacterial diversity of the native species *L. chinensis* was significantly higher compared to the three invasive leaf miner species. However, the diversity measurements did not differ significantly among the three invasive leaf miner species (Fig. 1A and Table S2). Principal coordinates analysis (PCoA) showed significant variation in microbiome composition among four leaf miner species (ADONIS: $R = 0.096$, $P < 0.001$), with PCoA1 (30.45%) and PCoA2 (13.83%) explaining 44.28% of the variation (Fig. 1B). All leaf miner species exhibited a high relative abundance of Proteobacteria and Actinobacteria, but all three invasive species were also associated with Firmicutes, while the native leaf miner, *L. chinensis*, instead harbored more Acidobacteriota (Fig. 1C). The endosymbiont *Wolbachia* (OTU9) was enriched in *L. huidobrensis* and present in the other two invasive leaf miners but not detected in the native *L. chinensis* (Fig. S2 and S3). While the microbiome composition varied among the four leaf miner species, this variation did not match host phylogeny (Fig. 1D).

Subsequently, network analysis showed that patterns of microbial co-occurrence differed among four leaf miner host species. We observed more connections in the bacterial communities from the three invasive species *L. trifolii* (288 edges, 274 positive and 14 negative), *L. sativae* (307 edges, 305 positive and 2 negative) and *L. huidobrensis* (804 edges, 752 positive and 52 negative) compared to the native *L. chinensis* (264 edges, 253 positive and 11 negative) (Fig. 1E and Table 1). The average path lengths were shorter, and diameters were smaller, in the networks of three leaf miner *L. huidobrensis*, *L. trifolii*, and *L. sativae* than in *L. chinensis* (Table 1), revealing closer relationships among each of the three invasive leaf miner enriched communities. Together, these results suggested that species co-occurred more frequently within the three invasive leaf miner microbial communities compared to that of the native *L. chinensis*.

**Comparison of bacterial communities in laboratory-reared versus wild leaf miners.** We compared the bacterial microbiota between laboratory-reared and wild leaf miners without considering the effects of the host species and geographical factors. The Shannon and richness indices of bacterial communities were 16.4% and 25.7% higher, respectively, in wild populations compared to laboratory-reared populations (Mann-Whitney U test: 1857, $P = 0.0158$) (Fig. S4A in the online supplemental material). Also, PCoA revealed significant differences in bacterial composition between wild and laboratory-reared *Liriomyza* (ADONIS: $R = 0.018$, $P < 0.001$) (Fig. S4B and C). The network complexity in laboratory-reared *Liriomyza* (Average degree:12.69) was higher than in the wild (Average degree:7.91). The number of edges was greater in laboratory-reared *Liriomyza* (425, 344 positive and 81 negative) than the found in wild flies (257, 257 positive) (Table S3).

**Multiple drivers of leaf miner bacterial communities.** We estimated the correlation between bacterial community variation and both host species and environmental variables (e.g., plant species, climatic and geographical factors) with a structural equation model (SEM). Results revealed that bacterial community composition was negatively

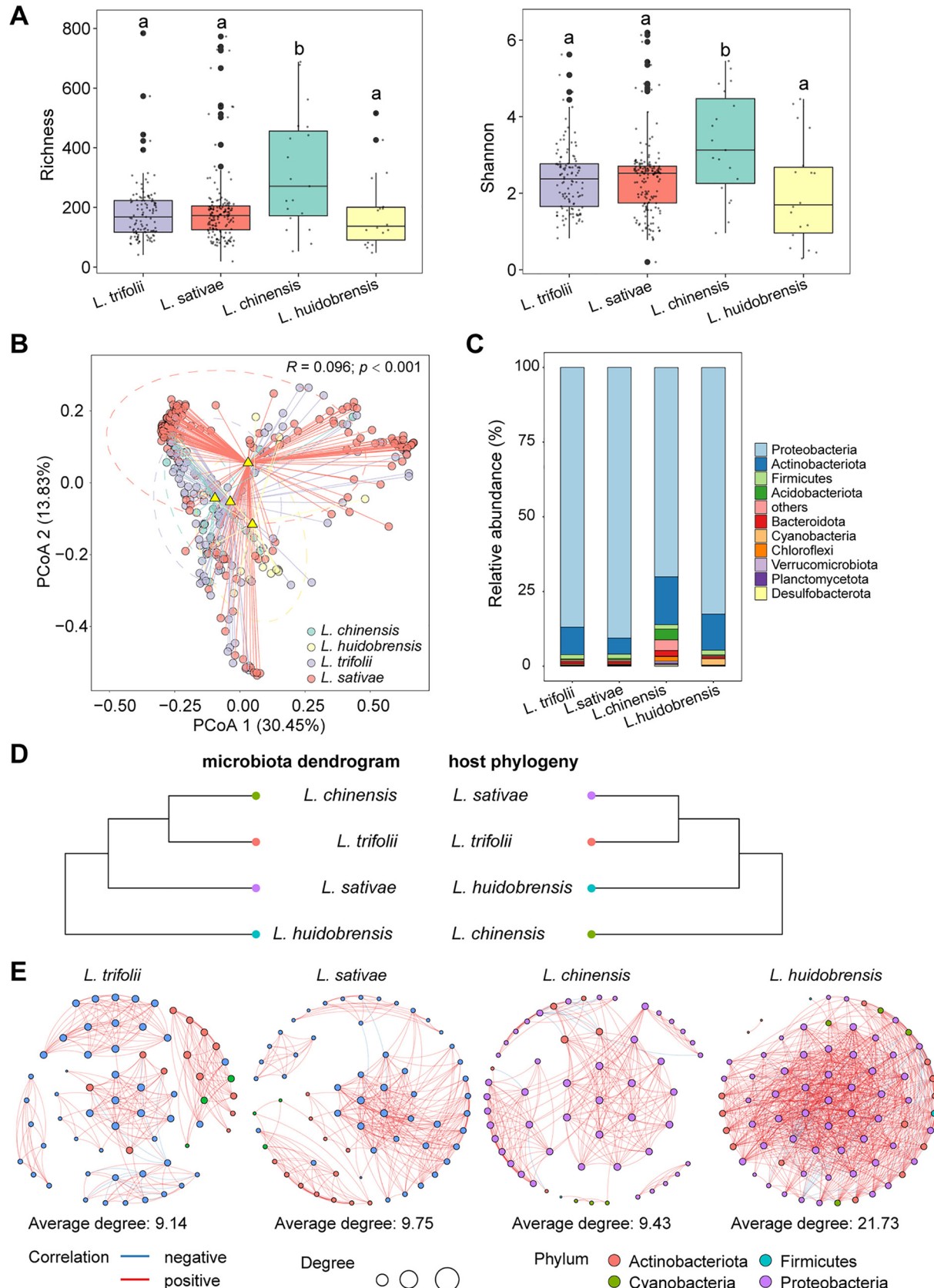

**FIG 1** Variation in bacterial community among four leaf miner species. (A) Shannon and Richness indices of four leaf miner bacterial communities. Different letters indicate significant differences between groups at a level of *P* < 0.05. (B) Principal coordinate analysis (PCoA)

**TABLE 1** Topological property of bacterial networks in each leaf miner species

| Group | Nodes | Edges | Positive edges | Negative edges | Connectance | Avg. degree | Avg. path length | Clustering coefficient | diameter |
|---|---|---|---|---|---|---|---|---|---|
| *L. trifolii* | 63 | 288 | 274 | 14 | 0.15 | 9.14 | 2.59 | 0.86 | 5.53 |
| *L. sativae* | 63 | 307 | 305 | 2 | 0.16 | 9.75 | 2.22 | 0.72 | 4.46 |
| *L. chinensis* | 56 | 264 | 253 | 11 | 0.17 | 9.43 | 3.79 | 0.83 | 6.93 |
| *L. huidobrensis* | 74 | 804 | 752 | 52 | 0.30 | 21.73 | 2.03 | 0.75 | 3.31 |

correlated with host species (path coefficients: -0.843) (Fig. 2). Leaf miner species was negatively associated with climatic factors (i.e., annual mean temperature and annual precipitation) and positively associated with food plant species (Fig. 2). These results suggested that host species was the most important driver in governing the bacterial compositions in the leaf miners. Environmental factors indirectly differentiate leaf miner bacterial community by changing the host attributes.

Redundancy analysis (RDA) showed that just 11.76% variance of the bacterial community could be explained by all examined factors, with RDA1 = 5%, RDA2 = 3%. However, >88% of the community variation could not be explained, implying complex processes of bacterial community assembly (Fig. S5 and Table S4 in the online supplemental material).

**Bacterial community assembly processes in leaf miners.** We explored the bacterial community assembly of leaf miners using neutral and null model analyses. The occurrence frequency of bacteria OTUs within the combined metacommunity of all four leaf miners (310 individuals) fit rather weakly to the neutral community model, and the majority of bacterial OTUs fell outside the 95% confidence interval for the neutral prediction ($R^2 = 0.641$; Fig. S6). For each leaf miner species analyzed separately, the degree of fit was also low in both invasive leaf miners *L. sativae* ($R^2 = 0.624$) and *L. trifolii* ($R^2 = 0.683$) was higher than that in native *L. chinensis* ($R^2 = 0.579$), with the lowest values in *L. huidobrensis* ($R^2 = 0.401$) (Fig. 3). This suggests that deterministic processes play a more critical role than stochasticity in the driving of leaf miner bacterial communities.

Null model analysis revealed that the bacterial community pooled across the four species was shaped primarily by variable selection and drift (Fig. S7 in the online supplemental material). Similar results were observed when the four leaf miner species were examined separately (Fig. 4A). However, the relative contribution of variable selection was higher for the native species *L. chinensis* (73.10%) than the three invasive species (*L. trifolii*: 63.18%, *L. huidobrensis*: 56.86% and *L. sativae*: 54.50%). In contrast, the relative contribution of drift was higher for three invasive species (*L. trifolii*: 27.24%, *L. huidobrensis*: 34.64% and *L. sativae*: 32.98%) relative to the native *L. chinensis* (24.56%) (Fig. 4B). Taken together, these results indicated that a combination of variable selection and drift drive the assembly of leaf miner bacterial communities, and their relative influence varies strongly with host species.

To evaluate the influence of environmental factors on leaf miner community assembly, we explored the relation between $\beta$NTI value and four environmental factors with the Mantel test. The results revealed that the $\beta$NTI was not significantly correlated with latitude ($R = -0.052$, $P = 0.976$), longitude ($R = -0.04$, $P = 0.906$), annual mean temperature ($R = -0.051$, $P = 0.955$) and annual mean precipitation ($R = -0.06$, $P = 0.986$) (Fig. 5), suggesting that the bacterial community assembly in leaf miners was not impacted by environmental factors.

**Potential sources of leaf miner microbes.** To determine sources of microbes that make up the leaf miner microbiota, we surveyed the microbiota associated with leaf

**FIG 1** Legend (Continued)

plot based on Bray-Curtis dissimilarities performed on the taxonomic profile for four leaf miner microbial communities. (C) The relative abundance of the major phyla presents in the microbial communities in the four leaf miner species. (D) Comparison of host phylogeny with dendrogram of similarity in microbiota composition in the four leaf miner species. (E) Co-occurrence networks of the microbiota in the four leaf miner species (i.e., *L. trifolii*, *L. sativae*, *L. chinensis*, *L. huidobrensis*). Edges represent significant Spearman correlations ($\rho > |0.6|$, $P < 0.05$). Blue and red lines represent significant negative and positive correlations, respectively. The sizes of the points indicate the relative abundance of OTUs in each microbial community.

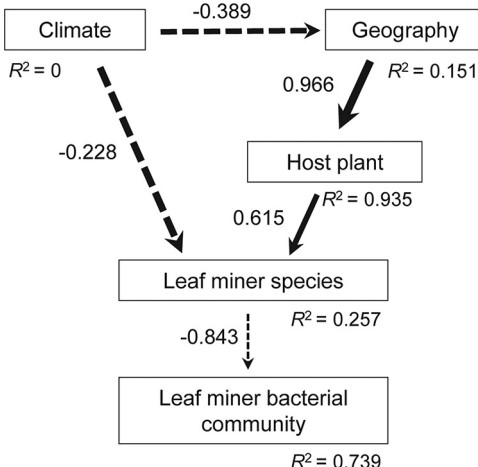

**FIG 2** The effect of multiple factors on the leaf miner bacterial community composition. Path diagram for the structural equation model (SEM) for environmental/host plant factors and microbial Bray-Curtis dissimilarity. The black solid and dashed arrows represent significant positive and negative paths, respectively. The width of the arrows represents the strengths of these relationships. The $R^2$ values under each box indicate the amount of variation in that variable explained by the input arrows. Numbers next to arrows are unstandardized slopes.

miner food plants and soil. Alpha diversity of bacterial communities, as indicated by the Shannon and richness indices, varied significantly among soil, root, leaf and leaf miners (Richness: $P < 0.0001$, $F = 33.45$; Shannon index: $P < 0.0001$, $F = 35.77$) (Fig. 6A). Bacterial diversity decreased from soil to root to plant to leaf miners (Fig. 6A). PCoA showed that larval and adult leaf miners microbiota formed a close cluster, which were distinct from those found in roots and soil samples (ADONIS: $R = 0.83$, $P < 0.001$) (Fig. 6B). Variation in bacterial community composition was also observed among leaf miners, leaves, roots and soil (Fig. 6C). There was no significant difference in alpha ($P > 0.05$) and beta diversity (Pair ADONIS: $R = 0.86$, $P = 0.22$) between larva and adult leaf miners (Fig. 6A and B). We then quantified the overlap of microbes among leaf miners, leaves, roots and soil. Just 368 OTUs (5.23%) were shared among all four groups (Fig. 6D). For leaf miner adults, the number of shared OTUs from adjacent trophic levels gradually decreased from leaves to roots to soil (Fig. 6D). The average degree of network complexity gradually increased from soils (15.65) to roots (24.94) and then to leaves (24.98) and was lowest in leaf miners (larva: 6.66; adult: 7.19) (Fig. 6E and Table 2).

Fast expectation-maximization microbial source tracking (FEAST) was conducted to identify potential sources of leaf miner-colonizing microbes. The results showed that the leaf miners acquired approximately 8.81% of their microbes from leaves, and 4.72% from roots and 1.38% from soil (Fig. 7). The large majority (85.4%) of leaf miner bacterial members did not derive from any of these (Fig. 7). Similar patterns were observed when considering just larval or adult flies (Fig. S8).

## DISCUSSION

In this study, we, for the first time, conducted an extensive and systematic investigation of the microbiomes of wild and lab-reared *Liriomyza* leaf miners and clarified the primary forces governing microbiome assembly and potential sources of leaf miner microbiota. Independent of host species and collection locality, we demonstrated that the microbiomes of wild leaf miners were more diverse than laboratory-reared populations. Similar results have been reported in some model organisms, such as the fruit fly *Drosophila* (40) and *Triatoma infestans* (49). The laboratory-reared microbial community has a much more connected and complex network structure compared to wild microbial communities. Of course, variability in the degree of selection or environmental factors under laboratory versus natural conditions likely alter microbiome assembly (40).

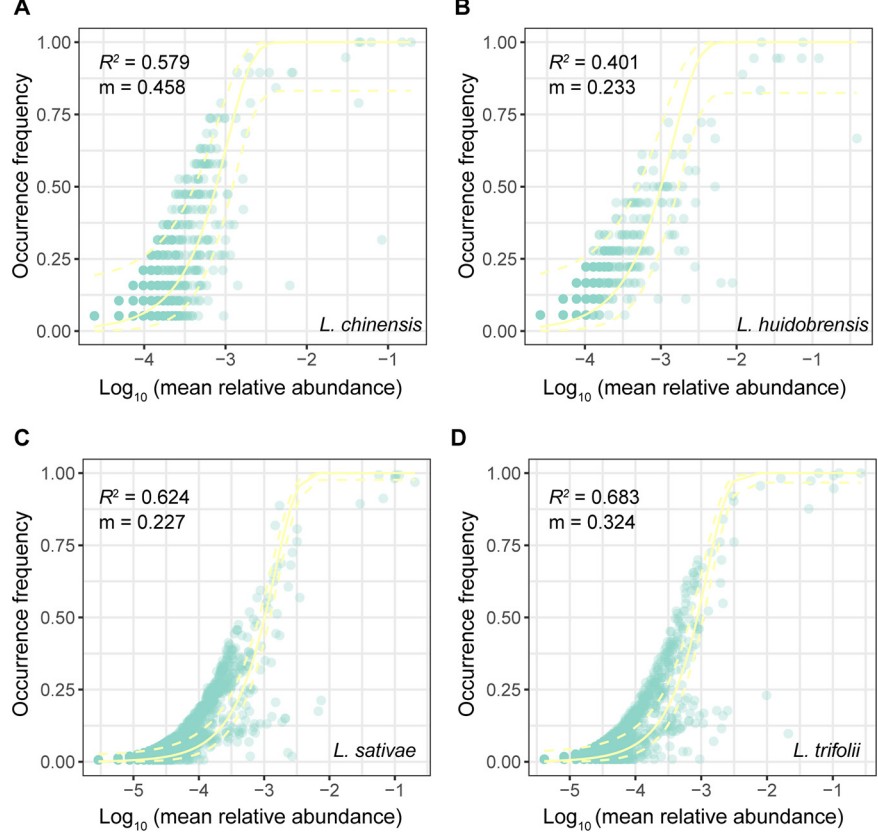

**FIG 3** Fit of a neutral model to each OTUs observed in four leaf miner species: (A) *L. chinensis*, (B) *L. huidobrensis*, (C) *L. sativae*, and (D) *L. trifolii*. The yellow solid and dashed lines represent the predicted occurrence frequency and 95% confidence interval of the neutral model, respectively. *m* indicates the estimated migration rate, and $R^2$ indicates the fit to the neutral model.

However, the relative importance of factors structuring microbiota assembly is still debated (50), and likely varies with system and conditions.

Our results in leaf-mining flies agree with the broadly observed pattern that host species identity is more generally more important than the environment in shaping the host-associated microbiota across diverse arthropods (51–53). The diversity and composition of bacterial communities differed significantly among the four leaf miner species examined. Remarkably, native leaf miner *L. chinensis* individuals harbored more diverse microbiotas than did those from the three invasive leaf miners (*L. trifolii*, *L. huidobrensis*, and *L. sativae*), among which microbial diversity did not differ significantly. One caveat, is that we examined only a single native leaf miner species, which is possibly not representative of native leaf miners. In contrast, the microbial community of the invasive species exhibited a more connected and complex network structure. We have several potential explanations for the observed variation in bacterial diversity and composition among native and invasive leaf miners. First, host physiochemical conditions and feeding habits vary among species, impacting the community composition of microbiota (26, 54, 55). The native species *L. chinensis* is oligophagous, while the three invasive *Liriomyza* species are highly polyphagous (45, 46). Several previous studies have demonstrated that host plant plays a substantial role in microbial diversity and composition (37, 56). Second, heritable endosymbionts potentially contribute to the structure of the environmentally acquired gut community. Our previous study supported that recent infection by *Wolbachia* alters microbial communities in wild *Laodelphax striatellus* populations (30). Similar results were reported for various systems, notably *Aedes aegypti* (57), *Armadillidium vulgare* (58), spider mites (22) and beetles (59). The impacts of *Wolbachia* on microbial communities may be common in natural systems, and potentially mediated

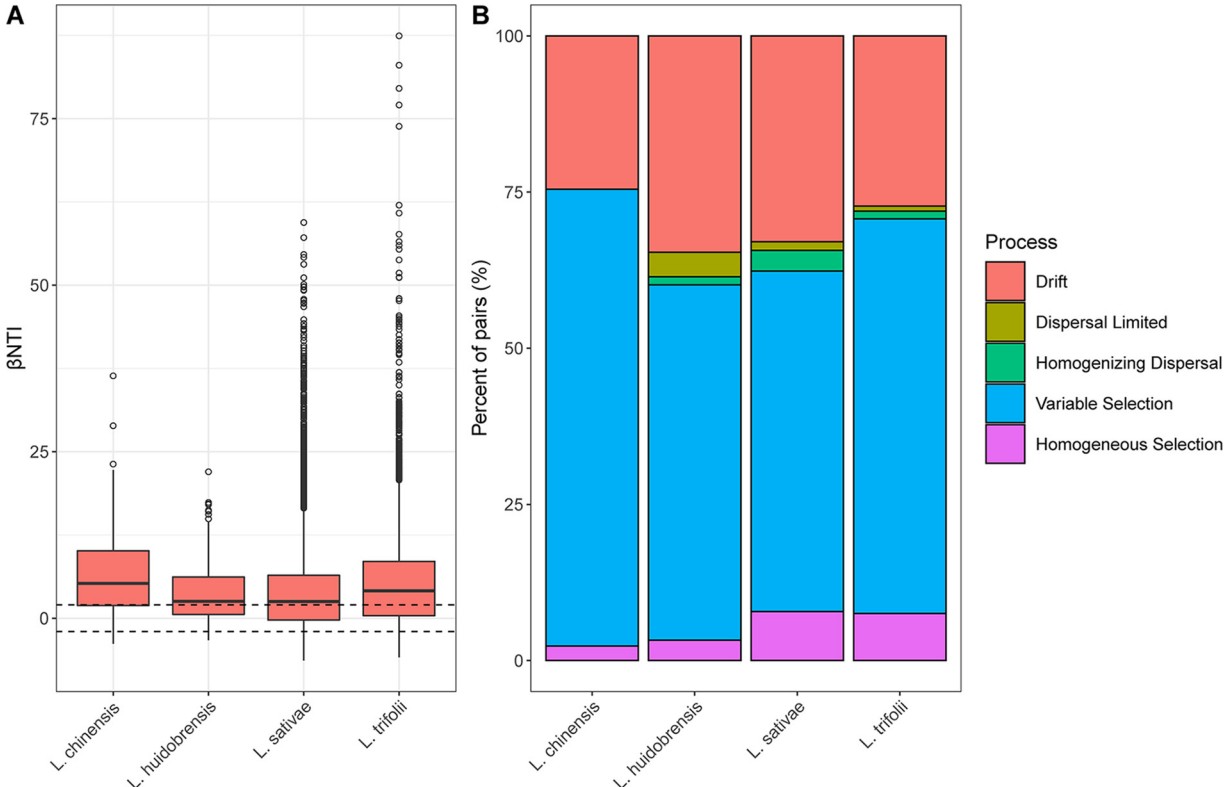

**FIG 4** Null model analysis revealing the assembly mechanism of the bacterial community in four leaf miner species. (A) Effects of deterministic ($\beta$NTI $\geq$ 2 or $\beta$NTI $\leq$ -2) and stochastic processes (-2 < $\beta$NTI < 2) in shaping the bacterial community in four leaf miner species. (B) The quantified major ecological process (i.e., homogeneous selection, heterogeneous selection, homogenizing dispersal, dispersal limited, and drift) in governing the microbial community of four leaf miner species.

by immune system modulation, resource competition or changes in metabolism (30, 57, 60). In our study, *Wolbachia* was detected in three leaf miner species, but not in the native species. In the invasive species *L. huidobrensis*, *Wolbachia* was the dominate bacteria taxa. More work is needed to determine whether and how *Wolbachia* impacts community composition in leaf-mining flies.

In most cases, deterministic and stochastic processes are not mutually exclusive, and instead work in conjunction in microbiome assembly (18, 19, 32). Yet analysis can estimate the relative contribution of these processes in shaping host microbiomes, which are likely to vary among systems. To illustrate, studies on gut microbial community assembly in zebrafish suggested homogeneous selection was the key factor (36), while stochastic forces (drift or stochastic dispersal) were relatively more important in fungal community assembly in leaves and roots in early sorghum development (61). Our results suggested that deterministic forces were more influential in shaping the microbiomes of leaf-mining flies. Of the ecological processes assessed, variable selection and drift were the major deterministic and stochastic processes driving the leaf miner bacterial community, respectively. In general, the degree of these processes governing microbiota assembly is associated with multiple factors (62). Furman et al. (37) showed stochasticity constrained by deterministic effects of diet and age drive rumen microbiome assembly dynamics. Variations in host and geography alter the relative contribution of different processes in assembling honeybee gut microbiota (41). In our study, the degree of these processes governing microbiota was correlated with host species but independent of either geographical or climatic factors, where the relative contribution of drift appeared higher in three invasive leaf miners than in native species. These results could explain the relatively higher fluctuation of the three invasive leaf miners bacterial communities compared to the native species. From a co-evolution

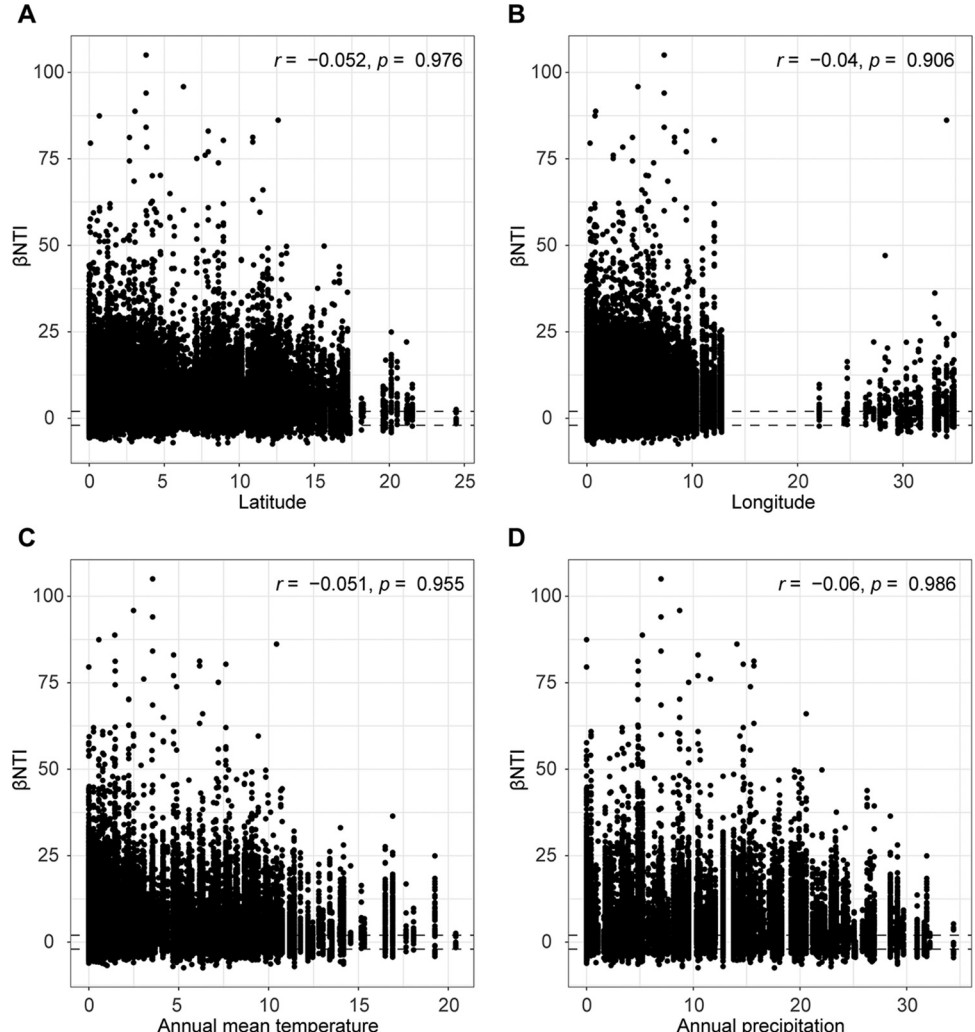

**FIG 5** Effects of environmental factors on leaf miner bacterial community assembly. Mantel analysis was used to evaluate the correlation between the $\beta$-Nearest Taxon Index ($\beta$NTI) and the multiples environmental variables: (A) latitude, (B) longitude, (C) annual mean temperature (AMT) and (D) annual mean precipitation (AP).

perspective, we agree with Ge et al. (41), who argue that the neutrality-based stochastic processes tend to be the main forces driving coevolution, and the deterministic process determines the direction of coevolution, which could partly explain why the microbiota relationships did not reflect host phylogeny in our study. However, this remains speculative in the absence of experimental data. Taken together, our results suggested that host species shapes the bacterial community in invasive leaf miners by changing the relative contribution of deterministic processes (i.e., variable selection) and stochastic (i.e., drift) processes (Fig. 8A).

Source tracking analysis revealed that the leaf miner *L. sativae* acquired a relatively higher proportion of their microbiomes from leaves than soil under natural conditions. Our findings appear to contrast with the recent work of Hannula et al. (44), who found that foliar-feeding caterpillars acquire microbes from the soil rather than the host plant. However, their study was performed under greenhouse conditions and the dynamics of microbiome acquisition may vary from those in natural settings. The discrepancy may also be due to biological traits differences between the two research systems. In leaf miners, both larvae and adults feed on plant leaves (45), while caterpillars can often have more frequent contact with the soil and adults are often nectar feeders (44). Hence, it is perhaps not surprising that these insects encounter different microbial source pools.

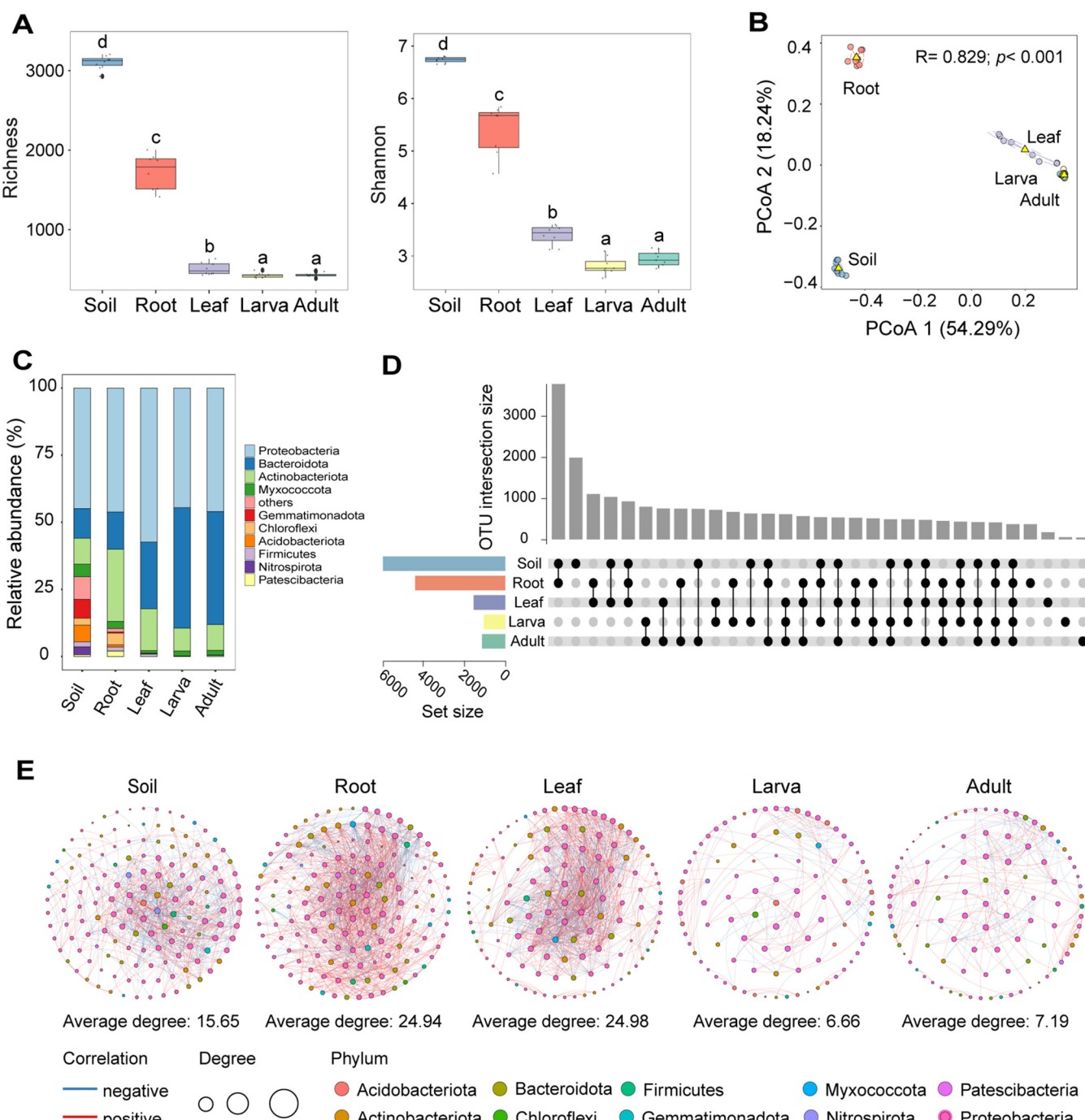

**FIG 6** Diversity and community structure of bacterial microbiota in leaf miners, leaves, roots and soil. (A) The alpha diversity, including Shannon and richness indices of the bacterial community in larva and adult leaf miners, leaves, roots and soil. Different letters denote significant differences between groups at a level of $P < 0.05$ based on Kruskal–Wallis test. (B) Principal coordinate analysis (PCoA) of bacterial community Bray-Curtis dissimilarities with *Adonis* test. The variation explained by the PCoA axes is given in parentheses. (C) Relative abundances of each bacterial phylum in different samples. (D) Setup diagram of shared and unique OTUs numbers observed in the larva and adult leaf miner, leaves, roots and soil. (E) Networks of the bacterial microbiota in the leaf miner, leaves, roots and soil. Edges represent significant Spearman correlations ($\rho > |0.6|$, $P < 0.05$). Light blue and red lines represent significant negative and positive correlations, respectively. The size of the points indicates the relative abundance of OTUs in each microbial community.

We created a visual model of the transmission dynamics of microbes among multiple trophic networks based on our analyses (Fig. 8B). This model showed that (i) bacterial diversity and network complexity decrease from soil to plants to leaf miners, and that (ii) leaf miners acquire more microbiota from adjacent trophic levels, with the potential for bacterial transfers decreasing along a continuum. Species interactions

**TABLE 2** Bacterial co-occurrence network characteristics in each host group

| Group | Nodes | Edges | Positive edges | Negative edges | Connectance | Avg. degree | Avg. path length | Clustering coefficient |
|---|---|---|---|---|---|---|---|---|
| Soil | 143 | 1119 | 587 | 532 | 0.11 | 15.65 | 2.75 | 0.55 |
| Root | 146 | 1819 | 1269 | 550 | 0.17 | 24.92 | 2.33 | 0.60 |
| Leaf | 125 | 1561 | 926 | 635 | 0.20 | 24.98 | 2.43 | 0.74 |
| Larva | 94 | 313 | 194 | 119 | 0.07 | 6.66 | 3.37 | 0.55 |
| Adult | 99 | 356 | 226 | 130 | 0.07 | 7.19 | 3.63 | 0.54 |

may influence these patterns. A recent study shows that insect herbivory reshapes a native leaf microbiome (63), raising the question of whether the insect microbiota may change the composition of the food plant or soil microbiome. Further study is needed to examine the direct and indirect interaction between soil, plant and insect microbiomes via labeled with isotopic tracers or other methods (44). Although microbiomes of soils and plants are linked, we still lack the understanding of the effect of this interaction on the aboveground herbivorous insect. Suppose the soil microbiomes will affect the microbiomes of insects feeding on plants that grow later in these soil through modification of the microbiome of their host plant (44). In this case, from the application perspective with the integrated system, we expect to perturb the balance of insect microbiota by alter the soil or plant microbiota, and therefore to controlling the pest.

In conclusion, our results provide a comprehensive overview of bacterial microbiota diversity and composition between a group of invasive and native leaf mining flies. We highlight that host species shape the bacterial community in leaf miners by altering the relative influence of community assembly processes. In addition, leaf miners acquire microbes primarily from their host plant rather than the soil. Our findings help to resolve the major modulators of microbiome assembly in animals, especially invasive insect species, in a changing world and also provide new insights into the microbial shift among the continuum associated with different systems. Deciphering the function of leaf miner-related microbiota in host invasion and adaption, as well as the impacts of invasions of insects and symbiotic microbes on ecosystems, is a vital area for future research.

## MATERIALS AND METHODS

**Collections and sample storage.** Four leaf miner species, *L. chinensis, L. trifolii, L. huidobrensis*, and *L. sativae*, were collected between 2014 and 2020 from six different host plants across 43 geographic locations in China (Fig. S1A and Table S1 in the online supplemental material). Climatic data (e.g., annual mean temperature and annual mean precipitation) for each sampling location were obtained from the Climate Data sets (https://psl.noaa.gov). Collected leaf miners were starved for 24 h to egest any food in their guts, and all samples were preserved in 100% ethanol and stored at −20°C until DNA extraction. Prior to DNA extraction, each sample was surface sterilized with 75% ethanol and sterile dH$_2$O.

To examine microbial movement among the soil-plant-leaf miner interaction under field conditions, we collected adult and larval leaf miners along with associated plant leaves, roots, and soil (Fig. S1B and Table S1 in the online supplemental material). In brief, eight individual bean plants were haphazardly

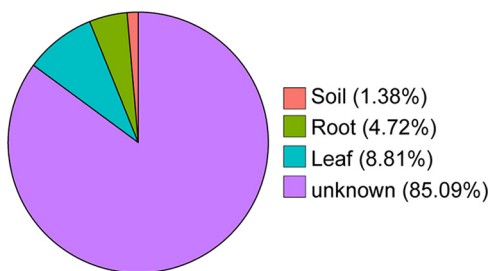

Soil (1.38%)
Root (4.72%)
Leaf (8.81%)
unknown (85.09%)

**FIG 7** Leaf miners acquired the bacterial community from food plants and soil. A pie shows the proportion of the potential sources of leaf miner microbes was derived primarily from leaves and gradually enriched by roots and soil.

**A**

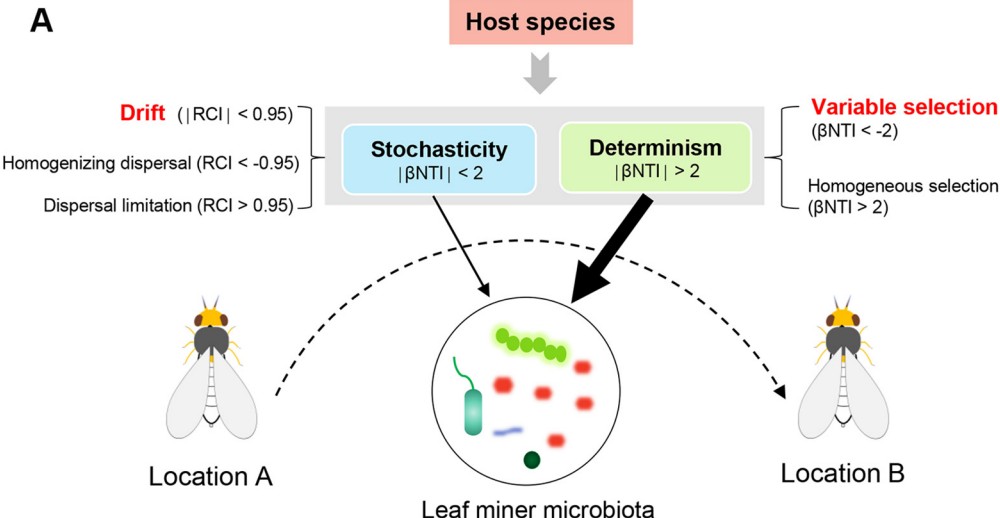

**B**

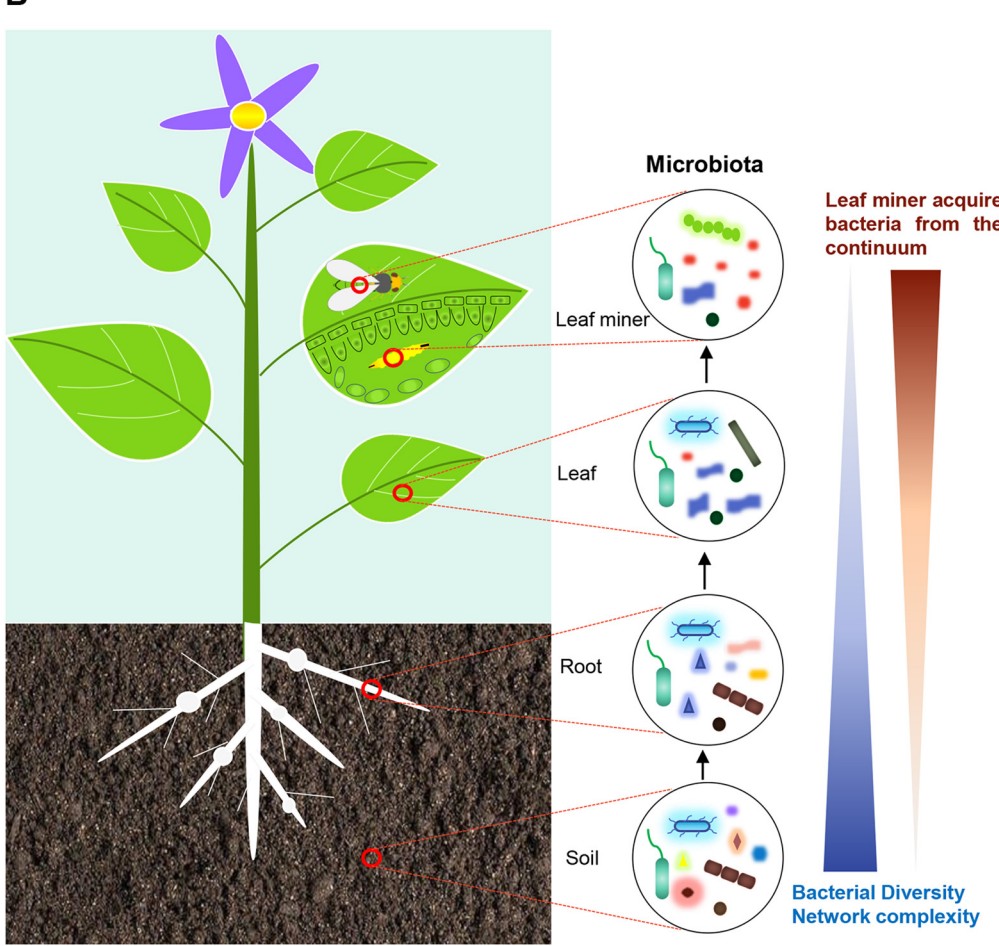

**FIG 8** A scenario for bacterial community assembly and shift in leaf miners. (A) Host species differentiate leaf miner bacterial communities by changing the relative contribution of community assembly processes during the leaf miner invade and spread. (B) Bacterial microbiota shift among the continuum of soil to plant to leaf miner. Bacterial diversity and network complexity tend to gradually reduce from the continuum of soil to plants to leaf miner. For leaf miners, the number of shared common OTUs from adjacent trophic levels gradually decreased from up to bottom.

selected, and one or two leaves with a *Liriomyza* mine from a mid-to-high position of the plant were clipped. Larvae were dissected from the leaf mine by a sterile needle, and a part of the same leaf without a mine was also collected for DNA extraction. Other leaves containing *Liriomyza* larvae were transferred to a 1.5 mL sterile centrifugal tube and kept under controlled conditions (26°C, 60% relative humidity, and 16: 8 h [L: D]) until the leaf miner eclosed as an adult. Soil and root material from the same plant were also collected afterwards. Soil was also obtained from around the root by taking the top 10 cm of soil and removing roots, stones and most macro-invertebrates using a sieve. All leaf miners (adults and larvae), leaves, roots and soil samples were stored at −20°C until processing.

Laboratory populations of the native *L. chinensis* were established on scallions (*Allium cepa* L.), while the three invasive leaf miners were established on bean plants (*Phaseolus vulgaris* L.). All were held at 26°C, 60% relative humidity, and 16: 8 h (L: D).

**DNA extraction, DNA library preparation and sequencing.** DNA was extracted from each whole leaf miner individual using DNeasy blood and tissue kit (Qiagen, Hilden, Germany) according to the manufacturer's protocols. For leaf, root and soil samples, DNA was extracted using DNeasy plant or PowerSoil Kit (Qiagen, Hilden, Germany). The quantity and quality of extracted DNA were evaluated by 1% agarose gel electrophoresis and an UV spectrophotometer (Nanodrop 2000), respectively.

The V3-V4 region of the 16S rRNA gene was amplified with the primers 341F (5′-CCTAYGGGRBGCASCAG-3′) and 806R (5′-GGACTACNNGGGTATCTAAT-3′) (64) to determine the bacterial composition of sampled material. PCR amplifications were performed in a total volume of 25 $\mu$L, using 12.5 $\mu$L 2× *Taq* Master Mix (Vazyme Biotech, China), 0.5 $\mu$L primer (20 $\mu$M each), and 1 $\mu$L of DNA, or ultrapure water for the PCR-negative controls. PCR conditions were set to 95°C for 5 min, followed by 27 cycles of 95°C for 30 s, 55°C for 30 s, and 72°C for 45 min, and a 72°C final extension for 10 min. The amplicons were extracted from 2% agarose gels and purified using the AxyPrep DNA Gel Extraction Kit (Axygen Biosciences, CA, USA) according to the manufacturer's instructions and quantified using QuantiFluorTM-ST (Promega, USA). Purified amplicons were pooled to reach equimolar concentration and paired-end sequenced (2 × 250 bp) on the Illumina HiSeq 2500 platform. All amplicon sequencing was performed by Shanghai Biozeron Biotechnology Co. Ltd (Shanghai, China). The raw sequence data were processed as described previously (24). Screened sequences were clustered into operational taxonomic units (OTUs) using UPARSE (65), with a similarity cutoff of 97%. The phylogenetic affiliation of each 16S rRNA gene sequence was analyzed with the RDP Classifier against the SILVA database (66).

**Bacterial community diversity analyses.** To determine the species richness and diversity of samples or groups, three alpha diversity indices, including richness, Pielou evenness and Shannon were calculated using the "*vegan*" R package (67). A nonparametric test was used to test for differences in microbiome variance among samples. The "*EasyStats*" R package was used to estimate and visualize a range of basic parameters that varied among samples. To evaluate differences in microbiome community composition among samples, we conducted a permutational multivariate analysis of variance (perMANOVA) analysis of Bray-Curtis dissimilarities. perMANOVA was performed using Adonis in '*vegan*'. Variation in bacterial composition among samples was visualized using PCoA. PCoA was performed using the R package "*vegan*" (67).

The phylogenetic relationship of four *Liriomyza* species mitochondrion genomes was generated by MEGA using a neighbor joining (NJ) method (68) based on the Poisson correction model with a bootstrap value of 1000. Microbiota composition dendrograms were carried out by averaging the microbiota composition of all samples for each species, followed by hierarchical clustering of Bray-Curtis dissimilarities by average linkage.

Network analyses were applied to reveal significant relationships between the relative abundance of OTUs using the sparse correlations for compositional data algorithm implemented in the SparCC python module (69). Robust correlations with Spearman's correlation coefficients > 0.6 and false discovery rate-corrected *P*-values < 0.05 were used to construct networks. To describe the topology of the networks, a set of metrics, including average degree, average path length, clustering coefficient, network diameter and centralization degree were calculated. To assess nonrandom patterns in the resulting network, we compared our network against its randomized version using the igraph package. The visualization was generated using the "*ggClusterNet*" R package.

**Environmental variables and bacterial community associations analysis.** To quantify the variation in leaf miner bacterial communities explained by various environmental variables, a SEM was employed on $\beta$-diversity data, using the '*sem*' function in the "*lavaan*" package in R (70).

Redundancy analysis (RDA) was used to find the contributions of each host-relation (e.g., leaf miner species and sex) or environmental factor (e.g., latitude, longitude, altitude and food plants) to the overall composition variation in the microbiome (71). RDA was carried out using the rda function in "*vegan*" package in R.

**Bacterial community assembly analyses.** Two approaches were used to infer the leaf miner bacterial community assembly process. Firstly, the Sloan neutral model was used to assess the relative contributions of stochastic processes to microbial community assembly (72). We identified OTUs that fell within, above or below the 95% confidence interval around the neutral model prediction as per Burns et al. (73). The 95% confidence interval was computed using the "*hmisc*" package in R. The goodness of fit of the neutral model to leaf miner data was assessed using the coefficient of determination ($R^2$). Secondly, a null model analysis was used to quality the relative contribution of ecological process (i.e., drift, selection, dispersal) in bacterial community assembly. We calculated the Raup-Crick index (RCI) and the beta Nearest Taxon Index ($\beta$NTI) using the null model in R (19). The $|\beta$NTI$| \geq 2$ and $|\beta$NTI$| < 2$ represent dominant deterministic processes and stochastic processes in shaping the microbial community, respectively. We then incorporated $\beta$NTI and RCI to estimate the relative strength of homogeneous selection ($\beta$NTI < −2),

variable selection ($\beta$NTI $>$ 2), homogeneous dispersal (RCI $<$ 0.95 and $|\beta$NTI$|$ $<$ 2), dispersal limitation (RCI $>$ 0.95 and $|\beta$NTI$|$ $<$ 2), and drift ($|$RCI$|$ $<$ 0.95 and $|\beta$NTI$|$ $<$ 2) in driving the composition of microbiota. The Mantel test was performed to evaluate the relationship between $\beta$NTI value and environmental variables.

**Source-tracking analysis.** To track the origin of the leaf miner microbiota, we used a source tracking method named fast expectation-maximization microbial source tracking (FEAST) on OTUs identified across the soil-root-leaf-leaf miner continuum (74). FEAST was conducted with the package "*FEAST*" in the R and visualized as a pie chart using the "*ggplot2*" package.

**Data availability.** Molecular sequence data have been deposited in the NCBI Sequence Read Archive (SRA) database (accession number PRJNA763753).

## SUPPLEMENTAL MATERIAL

Supplemental material is available online only.
**SUPPLEMENTAL FILE 1**, PDF file, 1 MB.
**SUPPLEMENTAL FILE 2**, XLSX file, 0.03 MB.

## ACKNOWLEDGMENTS

We are very grateful to Kerry M. Oliver of University of Georgia for his valuable comments and suggestions on the manuscript; Gao Hu and Le Huang of Nanjing Agricultural University for help with the climatic data collection; and Jing-Yun Chen and Xiao-Xiang Zhang of Yangzhou University, China, for collecting leaf miner samples. This work was supported by the National Natural Science Foundation of China (31901888), and the Jiangsu Science & Technology Support Program (BE2014410).

Y.X.Z. and Y.Z.D. designed the study; Y.X.Z., Y.W.C., R.Y., X.Y.W., M.X.L., and Y.C.W. performed the research; Y.X.Z. and T.W. conducted the statistical analyses; Y.X.Z. and Y.Z.D. wrote the manuscript. All authors read and approved the final manuscript.

We declare that we have no conflict of interest.

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
