## [Reviewer comments · Microbiology Spectrum]

Microbiology Spectrum

Species Identity Dominates over Environment in Driving Bacterial Community Assembly in Wild Invasive Leaf Miners

Yu-Xi Zhu, Ya-Wen Chang, Tao Wen, Run Yang, Yu-Cheng Wang, Xin-Yu Wang, Ming-Xing Lu, and Yu-Zhou Du

Corresponding Author(s): Yu-Zhou Du, Yangzhou University

Review Timeline:

Submission Date:	January 24, 2022
Editorial Decision:	February 19, 2022
Revision Received:	March 1, 2022
Accepted:	March 7, 2022

Editor: Daifeng Cheng

Reviewer(s): Disclosure of reviewer identity is with reference to reviewer comments included in decision letter(s). The following individuals involved in review of your submission have agreed to reveal their identity: Haijiang Huang (Reviewer #2)

Transaction Report:

DOI: <https://doi.org/10.1128/spectrum.00266-22>

February 19, 2022

Prof. Yu-Zhou Du
Yangzhou University
Yangzhou
China

Re: Spectrum00266-22 (Species Identity Dominates over Environment in Driving Bacterial Community Assembly in Wild Invasive Leaf Miners)

Dear Prof. Yu-Zhou Du:

Link Not Available

Sincerely,

Daifeng Cheng

Journals Department
Reviewer comments:

Reviewer #1 (Comments for the Author):

The assembly process of microbial communities is a long-standing interest in the field of microbial ecology. In particular, the driving forces shape intestinal microbiome are critical to decipher the host-microbe interactions. Herein, Zhu and co-authors worked on the microbiota of one native leaf miner *L. chinensis* and three close-related invasive species *L. trifolii*, *L. huidobrensis*, and *L. sativae*. By using structural equation models, they qualified host species identity was more important than environmental factors. Subsequently, they found variable selection was important to drive microbial community assembly through neutral and null model analysis. Further source tracking analysis showed that leaf miners might acquire microbes from host plants rather than the surrounding soil. In general, the work represents some interesting and informative results to the field, experimental

design was carefully carried out, and Materials & Methods of the article are described in detail. However, several concerns needed to be addressed before publication.

Major issues:

1. In Fig.1B, according to the illustration, grey represents *L. trifolii*. However, according to the figure, purple refers to *L. trifolii*.
2. "Reads for the endosymbiont *Wolbachia* (OTU9) were enriched in *L. huidobrensis* and present in the other two invasive leaf miners but not recovered from the native *L. chinensis* (Supplementary Fig. 2 and 3)". I didn't see the relevance between *Wolbachia* and gut microbiome. More details need to be added.
3. Fig.2 suggested that host species was the most important driver in governing the bacterial compositions in the leaf miners. However, to me, it is still unclear that the correlation between climate (geography/host plant) and leaf miner bacterial communities. Need to be clear somehow.
4. "PCoA showed that larval and adult leaf miners microbiota formed a close cluster, which were distinct from those found in leaves, roots and soil samples (ADONIS: $R = 0.83$, $p < 0.001$) (Fig. 6B)." However, given larvae, adults and leaves are close in Fig 6B, it's unfair to say 'distinct from those found in leaf'.
5. There are many spelling errors throughout paper, please doublecheck accordingly. I listed few of them below.
6. Fig. 3 B - D, these pictures are inconsistent with figure legends (line 735).
7. Fig. 6 A and D, it is more harmonious to keep the same group colors in the two images.

Minor issue:

1. The results of abstracts should be past tense.
2. Line 150, "thre" should be "three".
3. Line 265, "the the", delete one.
4. Line 274, "wer" should be "were".
5. Line 329, "indicate" should be "indicated".
6. Line 408, "analyses" should be "analysis".
7. Line 411, "was they" should be "is the".
8. Line 419, "show" should be "showed".
9. Line 433, "suggests" should be "suggested".
10. Line 448, "show" should be "showed".
11. In line 466, "...and native leaf leaf mining flies..." word "leaf" is repeated.
12. The page number connectors (-) of references are inconsistent. Such as lines 517, 523, 526...
13. The number of author names in the fifth reference is incorrect.

Reviewer #2 (Comments for the Author):

The manuscript entitled "Species identity dominates over environment in driving bacterial community assembly in wild invasive leaf miners" present findings from a survey of the bacterial communities associated with four leaf miner species, totaling 310 individuals across 43 geographical populations in China, and the continuum of soil - plants - leaf miner. The findings are interesting in that they appear to be relatively novel in the scope of animal microbiome, and provide insights into the patterns of bacterial assembly and microbial source tracking. Main conclusion in the study is that species identity dominates over environment in driving bacterial community assembly in wild leaf miners.

This is a really great study and I'm happy the authors are reporting it.

Methods need a bit of clarification. I have a couple concerns about the method associated with bioinformatics analysis that can be cleared up relatively easily. Reviewers may want more detail on the bioinformatics pipelines for data filtering and taxon assignment.

A 'IMPORTANCE' Section is required.

Specific Comments:

Line 151: A space is required to add between 'at' and '26{degree sign}C'.

Line 155: These are whole insect extractions, rather than just gut? The author should clarify the methods they used.

Line 164: Did you also run 'blank' samples as negative controls?

Line 172: There is no mention of the how the library prep was done.

Line 178: Citation for this?

Line 182: Were analyses performed on rarefied samples or was another method of sequence depth standardization used?

Line 188: perMANOVA

Line 222-239: The rationale for the neutral and null models appears to be similar. Many readers will not be familiar the distinctions because they are similar so I suggest providing clear reasons for doing each model. This same thing applies to most of the other models, I would put into easily understandable language the reason you run each of these models/tests.

Line 311: Please mention the name of the neutral model you use and briefly explain it already here.

Line 465: This sounded like you provided a comprehensive overview of all leaf mining flies which I think number in the thousands.

The discussion section is thorough and interesting.

Fig. 1-Bacterial taxonomy at the phylum level is not particularly informative. Figures at the class level could provide additional insights into potential functional groups present in the leaf miners.

Fig. 2- This is a good figure.

Staff Comments:

Preparing Revision Guidelines

Please return the manuscript within 60 days; if you cannot complete the modification within this time period, please contact me. If you do not wish to modify the manuscript and prefer to submit it to another journal, please notify me of your decision immediately so that the manuscript may be formally withdrawn from consideration by Microbiology Spectrum.

Response to Reviewers

March 2, 2022

Manuscript ID: Spectrum00266-22

Title: Species Identity Dominates over Environment in Driving Bacterial Community Assembly in Wild Invasive Leaf Miners

Dear Daifeng Cheng,

Please find enclosed our revised manuscript entitled '**Species Identity Dominates over Environment in Driving Bacterial Community Assembly in Wild Invasive Leaf Miners**'. The comments from the reviewers were very constructive and have helped us improve the manuscript. Below, we list our responses point by point corresponding to the comments made by the reviewers.

Sincerely yours,

Yu-Zhou Du, Professor

School of Horticulture and Plant Protection & Institute of Applied Entomology

Yangzhou University

Yangzhou, Jiangsu 225009

CHINA

E-mail: yzdu@yzu.edu.cn

Reviewer comments:

Reviewer #1 (Comments for the Author):

The assembly process of microbial communities is a long-standing interest in the field of microbial ecology. In particular, the driving forces shape intestinal microbiome are critical to decipher the host-microbe interactions. Herein, Zhu and co-authors worked on the microbiota of one native leaf miner *L. chinensis* and three close-related invasive species *L. trifolii*, *L. huidobrensis*, and *L. sativae*. By using structural equation models, they qualified host species identity was more important than environmental factors. Subsequently, they found variable selection was important to drive microbial community assembly through neutral and null model analysis. Further source tracking analysis showed that leaf miners might acquire microbes from host plants rather than the surrounding soil. In general, the work represents some interesting and informative results to the field, experimental design was carefully carried out, and Materials & Methods of the article are described in detail. However, several concerns needed to be addressed before publication.

Response: We thank you for your helpful comments on the manuscript.

Major issues:

Point 1: In Fig.1B, according to the illustration, grey represents *L. trifolii*. However, according to the figure, purple refers to *L. trifolii*.

Response: We corrected Fig. 1B in the revised version.

Point 2: "Reads for the endosymbiont *Wolbachia* (OTU9) were enriched in *L. huidobrensis* and present in the other two invasive leaf miners but not recovered from the native *L. chinensis* (Supplementary Fig. 2 and 3)". I didn't see the relevance between *Wolbachia* and gut microbiome. More details need to be added.

Response: We removed Fig. S3 that showed the relevance between *Wolbachia* and other bacteria. However, this figure was irrelevant to our stated results here. Instead, we added a new figure showing the leaf miner bacterial communities at the genus

level.

Point 3: Fig.2 suggested that host species was the most important driver in governing the bacterial compositions in the leaf miners. However, to me, it is still unclear that the correlation between climate (geography/host plant) and leaf miner bacterial communities. Need to be clear somehow.

Response: As shown in Fig. 2, SEM analysis revealed that there was no significant correlation between climatic factors and leaf miner microbiota. Environmental factors (e.g., food plant species, climatic and geographical factors) indirectly differentiate leaf miner bacterial community by changing the host attributes. In fact, we stated the finding in line 179-181. Please check it.

Point 4: "PCoA showed that larval and adult leaf miners microbiota formed a close cluster, which were distinct from those found in leaves, roots and soil samples (ADONIS: $R = 0.83$, $p < 0.001$) (Fig. 6B)." However, given larvae, adults and leaves are close in Fig 6B, it's unfair to say 'distinct from those found in leaf'.

Response: To describe this correctly, we changed the sentence into 'PCoA showed that larval and adult leaf miners microbiota formed a close cluster, which were distinct from those found in roots and soil samples (ADONIS: $R = 0.83$, $p < 0.001$) (Fig. 6B)' in line 220-221.

Point 5: There are many spelling errors throughout paper, please doublecheck accordingly. I listed few of them below.

Response: Thanks for your helpful suggestions. We corrected those spelling errors and checked the spelling carefully throughout the manuscript in the revised version.

Point 6: Fig. 3 B - D, these pictures are inconsistent with figure legends (line 735).

Response: We are so sorry to make a mistake here. We have corrected figure legends in the revision.

Point 7: Fig. 6 A and D, it is more harmonious to keep the same group colors in the

two images.

Response: Thanks for your suggestions. To make sure the same group colors in the two images, we changed Fig. 6D.

Minor issue:

Point 1: The results of abstracts should be past tense.

Response: We corrected the tense error in the abstract and checked it elsewhere.

Point 2: Line 150, "thre" should be "three".

Response: We changed 'thre' into 'three' in line 154.

Point 3: Line 265, "the the", delete one.

Response: Done.

Point 4: Line 274, "wer" should be "were".

Response: We corrected the spelling error in our revision.

Point 5: Line 329, "indicate" should be "indicated".

Response: We changed 'indicate' into 'indicated' in line xx.

Point 6: Line 408, "analyses" should be "analysis".

Response: We changed 'analyses' into 'analysis' in line 205.

Point 7: Line 411, "was they" should be "is the".

Response: Done.

Point 8: Line 419, "show" should be "showed".

Response: We corrected it in our revision.

Point 9: Line 433, "suggests" should be "suggested".

Response: We changed 'suggests' into 'suggested' in line 304.

Point 10: Line 448, "show" should be "showed".

Response: We changed 'show' into 'showed' in line 319.

Point 11: In line 466, "...and native leaf leaf mining flies..." word "leaf" is repeated.

Response: One of doubled words was deleted.

Point 12: The page number connectors (-) of references are inconsistent. Such as lines 517, 523, 526...

Response: We corrected the page number connectors throughout the manuscript.

Point 13: The number of author names in the fifth reference is incorrect.

Response: We checked references carefully throughout the MS in the revised version.

Reviewer #2 (Comments for the Author):

The manuscript entitled "Species identity dominates over environment in driving bacterial community assembly in wild invasive leaf miners" present findings from a survey of the bacterial communities associated with four leaf miner species, totaling 310 individuals across 43 geographical populations in China, and the continuum of soil - plants - leaf miner. The findings are interesting in that they appear to be relatively novel in the scope of animal microbiome, and provide insights into the patterns of bacterial assembly and microbial source tracking. Main conclusion in the study is that species identity dominates over environment in driving bacterial community assembly in wild leaf miners.

This is a really great study and I'm happy the authors are reporting it.

Response: We thank you for your helpful comments on the manuscript.

Point 1: Methods need a bit of clarification. I have a couple concerns about the

method associated with bioinformatics analysis that can be cleared up relatively easily. Reviewers may want more detail on the bioinformatics pipelines for data filtering and taxon assignment.

Response: To clarify our methods, we added the sentences in line 391-392: The raw sequence data were processed as described previously (24).

In line 392-395, we stated the bioinformatics pipelines for taxon assignment. Please check it.

Point 2: A 'IMPORTANCE' Section is required.

Response: We added the following section in line 42-52:

IMPORTANCE

The invasion of foreign species, including leaf miners, is a major threat to world biota. Host-associated microbiota may facilitate host adaptation and expansion in a variety of ways. Thus, understanding the processes that drive leaf miner microbiota assembly is imperative for better management of invasive species. However, how microbial communities assemble during the leaf miner invasions, and how predictable the processes remain unexplored. This work quantitatively deciphers the relative importance of deterministic process and stochastic process in governing the assembly of four leaf miner microbiotas and identifies potential sources of leaf miner-colonizing microbes from the soil-plant-leaf miner continuum. Our study provides new insights into the mechanisms underlying the drive of leaf miner microbiota assembly.

Specific Comments:

Point 3: Line 151: A space is required to add between 'at' and '26°C'.

Response: Done.

Point 4: Line 155: These are whole insect extractions, rather than just gut? The author should clarify the methods they used.

Response: To make this clearer, we changed the sentence into "DNA was extracted from each whole leaf miner individual using DNeasy blood and tissue kit (Qiagen,

Hilden, Germany) according to the manufacturer's protocols." in line 373-375.

Point 5: Line 164: Did you also run 'blank' samples as negative controls?

Response: Yes, the negative control was used for PCR amplification.

Point 6: Line 172: There is no mention of the how the library prep was done.

Response: In fact, we stated the library prep in line 386-390. Please have a check.

Point 7: Line 178: Citation for this?

Response: We added the references in line 701-704: '66. McDonald D, Price MN, Goodrich J, Nawrocki EP, DeSantis TZ, Probst A, Andersen GL, Knight R, Hugenholtz P. 2012. An improved Greengenes taxonomy with explicit ranks for ecological and evolutionary analyses of bacteria and archaea. *ISME J* 6:610-618. [https://doi.org/10.1038/ismej.2011.139.](https://doi.org/10.1038/ismej.2011.139)'

Point 8: Line 182: Were analyses performed on rarefied samples or was another method of sequence depth standardization used?

Response: Bacterial diversity analyses were performed with the method of sequence depth standardization.

Point 9: Line 188: perMANOVA

Response: We changed "PERMANOVA" into "perMANOVA" in line xx.

Point 10: Line 222-239: The rationale for the neutral and null models appears to be similar. Many readers will not be familiar the distinctions because they are similar so I suggest providing clear reasons for doing each model. This same thing applies to most of the other models, I would put into easily understandable language the reason you run each of these models/tests.

Response: A neutral model was used to assess the relative contributions of stochastic processes to microbial community assembly, while a null model analysis was performed to quantify the relative contribution of ecological process (i.e. drift, selection,

dispersal) in bacterial community assembly. A combination of the two approaches can reveal the mechanism of microbial community assembly. In fact, we explained the distinctions between two approaches in our manuscript (line 431-440).

Point 11: Line 311: Please mention the name of the neutral model you use and briefly explain it already here.

Response: We added 'Sloan' in line 432.

Point 12: Line 465: This sounded like you provided a comprehensive overview of all leaf mining flies which I think number in the thousands.

Response: In our study, a total of 310 leaf miner individuals were used to investigate their microbiota. Thus, we argued that our study provided a comprehensive overview of bacterial microbiota diversity and composition of four leaf miner species.

Point 13: The discussion section is thorough and interesting.

Response: Thanks.

Point 14: Fig. 1-Bacterial taxonomy at the phylum level is not particularly informative. Figures at the class level could provide additional insights into potential functional groups present in the leaf miners.

Response: We added Figure S3 in the revised manuscript, which illustrates the leaf miner bacterial communities at the genus level.

Point 15: Fig. 2- This is a good figure.

Response: Thanks.

March 7, 2022

Prof. Yu-Zhou Du
Yangzhou University
Yangzhou
China

Re: Spectrum00266-22R1 (Species Identity Dominates over Environment in Driving Bacterial Community Assembly in Wild Invasive Leaf Miners)

Dear Prof. Yu-Zhou Du:

Your manuscript has been accepted, and I am forwarding it to the ASM Journals Department for publication. You will be notified when your proofs are ready to be viewed.

Sincerely,

Daifeng Cheng
Editor, Microbiology Spectrum

Journals Department
Supplemental Material: Accept
Supplemental Material: Accept